# N-SREBP2 Provides a Mechanism for Dynamic Control of Cellular Cholesterol Homeostasis

**DOI:** 10.3390/cells13151255

**Published:** 2024-07-25

**Authors:** Tozen Ozkan-Nikitaras, Dominika J. Grzesik, Lisa E. L. Romano, J. P. Chapple, Peter J. King, Carol C. Shoulders

**Affiliations:** 1Centre for Endocrinology, William Harvey Research Institute, Barts and The London School of Medicine and Dentistry, Queen Mary University of London, London EC1M 6BQ, UK; tozen_5@hotmail.com (T.O.-N.); dominika.grzesik@gu.se (D.J.G.); lisa.romano@ufl.edu (L.E.L.R.); j.p.chapple@qmul.ac.uk (J.P.C.); p.j.king@qmul.ac.uk (P.J.K.); 2Department of Chemistry and Molecular Biology, University of Gothenburg, 405 30 Göteborg, Sweden; 3Wallenberg Centre for Molecular and Translational Medicine, University of Gothenburg, 405 30 Göteborg, Sweden

**Keywords:** cholesterol, nucleus, control, ER

## Abstract

Cholesterol is required to maintain the functional integrity of cellular membrane systems and signalling pathways, but its supply must be closely and dynamically regulated because excess cholesterol is toxic. Sterol regulatory element-binding protein 2 (SREBP2) and the ER-resident protein HMG-CoA reductase (HMGCR) are key regulators of cholesterol biosynthesis. Here, we assessed the mechanistic aspects of their regulation in hepatic cells. Unexpectedly, we found that the transcriptionally active fragment of SREBP2 (N-SREBP2) was produced constitutively. Moreover, in the absence of an exogenous cholesterol supply, nuclear N-SREBP2 became resistant to proteasome-mediated degradation. This resistance was paired with increased occupancy at the *HMGCR* promoter and *HMGCR* expression. Inhibiting nuclear N-SREBP2 degradation did not increase *HMGCR* RNA levels; this increase required cholesterol depletion. Our findings, combined with previous physiological and biophysical investigations, suggest a new model of SREBP2-mediated regulation of cholesterol biosynthesis in the organ that handles large and rapid fluctuations in the dietary supply of this key lipid. Specifically, in the nucleus, cholesterol and the ubiquitin–proteasome system provide a short-loop system that modulates the rate of cholesterol biosynthesis via regulation of nuclear N-SREBP2 turnover and *HMGCR* expression. Our findings have important implications for maintaining cellular cholesterol homeostasis and lowering blood cholesterol via the SREBP2-HMGCR axis.

## 1. Introduction

Cholesterol is an integral component of lipid membranes, modulating their thickness, rigidity, and permeability and a range of transmembrane signalling pathways [1,2,3,4]. It is also used to promote the delivery of a diverse range of proteins from the ER to the Golgi apparatus en route to their final destinations [5,6], the integrity of myelin sheaths [7], and the synthesis of steroid hormones [8] and bile acids [9]. Hence, defects in tissue and cellular cholesterol homeostasis manifest in diverse diseases, including neurological diseases such as Alzheimer’s and Huntingdon’s Diseases [7,10], osteoporosis [11], non-alcoholic fatty liver disease [12], and coronary artery disease [13], commonly caused by the impaired hepatic uptake of LDL-C [14]. Less well appreciated is that inborn errors of cholesterol biosynthesis cause multi-system malformation syndromes, including Smith–Lemli–Opitz syndrome and Hydrops-ectopic calcification-moth-eaten skeletal dysplasia [15]. Moreover, our daily cholesterol requirements derive mainly from de novo synthesis [16,17].

The cholesterol-lowering effects of orthotopic liver transplantation in homozygous Familial Hypercholesterolemia patients [14] and the hepatic-specific knock-down of genes required for cholesterol synthesis in mice [18,19] amply demonstrate the central role of this organ in regulating blood cholesterol levels. Yet, the mechanistic understanding of the processes regulating hepatic de novo cholesterol synthesis, including those mediated by sterol regulatory element-binding protein (SREBP) 2, is predominantly extrapolated from studies performed in extrahepatic cells [20,21,22,23,24,25] that do not model key aspects of the hepatocyte’s functions. These include the assembly of VLDL for the bulk secretion of cholesterol, triglycerides, and cholesteryl esters into the circulation for delivery to peripheral cells [26]. Thus, both VLDL and the product of VLDL catabolism, LDL-C, are crucial for the body’s lipid economy, supplying extrahepatic cells with physiologically important substrates, thereby affecting their functions and dysfunction [27,28].

SREBP2, a member of the basic helix-loop-helix leucine zipper (bHLHZ) family of transcription factors, promotes cholesterol biosynthesis, whereas SREBP1 preferentially regulates the expression of genes involved in fatty acid synthesis [29,30,31,32]. Both are synthesised as inactive ER-membrane-bound precursors [33] and transported to the Golgi apparatus. Here, their soluble C-terminal sequences (C-SREBP) are released via the site 1-protease, a pre-requisite for the release of their soluble N-terminal transcriptionally active fragment (N-SREBP) via the site 2-protease [20,34,35,36]. In mice, transgenic expression of this soluble N-SREBP2 fragment markedly increases hepatic levels of mRNAs encoding enzymes in the cholesterol biosynthesis pathway and the actual rate of hepatic de novo cholesterol synthesis [29,31].

Studies performed in CHO-K1 and HeLa cells indicate that low ER membrane cholesterol triggers the transport of SREBP2 from the ER to the Golgi apparatus for proteolytic cleavage [37]. Similar to SREBP2, HMGCR is an intrinsic ER membrane protein [38] whose activity is regulated by changes in cholesterol flux [39,40,41], prompting the question of whether SREBP2 and HMGCR operate independently or in coordination to regulate hepatic cholesterol homeostasis. Furthermore, would a deeper, more integrated understanding of the quantitative, spatial, and temporal aspects of N-SREBP2 production and HMGCR regulation in cells that regulate blood cholesterol levels challenge specific aspects of the prevailing model of SREBP2 activation by cholesterol depletion [37,42]?

## 2. Materials and Methods

### 2.1. Cell Culture

McArdle-RH7777 (CRL 1601) and HepG2 cells (HB-8065™) were obtained from ATCC. Monolayers of cells were cultured in high-glucose (4500 mg/L) DMEM supplemented with 10% Fetal Bovine Serum (Invitrogen, Paisley, Renfrewshire, UK) or 5% lipoprotein-deficient serum (Sigma, Saint Louis, MO, USA), 3 mM L-glutamine (Gibco^TM^), and 150 U/mL penicillin/150 mg/mL streptomycin (Sigma, Waltham, MA USA) at 37 °C with 5% CO_2_. Atorvastatin (calcium salt) was purchased from Sigma.

### 2.2. Antibodies

The following antibodies were used in this study: anti-C-SREBP2 (Proteintech, Manchester, UK, 14508-1-AP, immunogen SREBP2 amino acid residues 794-1141; WB and IF, 1:750), anti-HMGCR (Abcam, Cambridge, UK, ab174830; WB, 1:1000), anti-N-SREBP2 (Abcam, ab30682, immunogen SREBP2 amino acids 455-469; WB and IF, 1:500), anti-N-SREBP1 (Abcam, ab3259, amino acids 301-407; WB and IF, 1:500), anti-GAPDH (R and D Systems, Abingdon, UK, WB, 1:10,000), monoclonal anti- β-actin (Sigma Aldrich, St. Louis, MO, USA, A5441; WB, 1:10,000), anti-TATA-box binding protein (Abcam, ab818; WB, 1:500), anti-KDEL (Enzo, Farmingdale, NY, USA, ADI-SPA-827-D; IF, 1:200), anti-Giantin (Abcam, ab37266; IF, 1:250), anti-PML (Abcam, ab96051; IF 1:200), and Anti-Ubiquitin antibodies (Abcam, ab7254; WB, 1:100). Secondary antibodies were Goat anti-Mouse IgG (LICOR, Lincoln, NE, USA, IRDye^®^ 800CW; WB, 1:10,000); Goat anti-Rabbit IgG (LICOR, IRDye^®^ 680CW, WB, 1:10,000); Goat anti-Rabbit IgG (H + L) Cross-Adsorbed Secondary Antibody, Alexa Fluor 488 (Invitrogen, A-11008; IF, 1:1000), and Goat anti-Mouse IgG (H + L) Cross-Adsorbed Secondary Antibody, Alexa Fluor 568 (Invitrogen, A-11004; IF, 1:1000).

### 2.3. Immunoblot Analyses

Total cell lysates were prepared by lysing cells in radioimmune precipitation assay (RIPA) buffer (Sigma). To prepare nuclear extracts, cells were incubated in ice-cold 10 mM HEPES pH7.9, 10 mM KCl and 1.5 mM MgCl_2_ for 10 min and homogenised in a Uniform Dounce Tissue Grinder (Jencons, East Grinstead, West Sussex, UK). Cytoplasmic fractions were removed by centrifugation at 1000× *g* for 15 min at 4 °C. Cell pellets (nuclear fractions) were washed with PBS, re-suspended in ice-cold 50 mM Tris-HCl pH7.5 and 1% SDS, and solubilised on ice via multiple 45 s bursts of sonication (Sonics & Materials, Inc., Newtown, CT, USA). Buffers were supplemented with cOmplete^TM^, EDTA-free Protease Inhibitor Cocktail (Sigma-Aldrich). Proteins were mixed with Laemmli sample buffer (Sigma) for size fractionation on 4–12% gradient polyacrylamide NuPage Bis-Tris precast gels (Invitrogen). Proteins were transferred to nitrocellulose membranes (Protran BA85, Whatman, Life Technologies, Carlsbad, CA, USA) and incubated with blocking buffer (5% non-fat milk powder in 1xPBS plus 0.4% Tween-20) for 1 h at RT. Incubations with primary antibodies in blocking buffer were performed overnight at 4 °C. Immunoreactive products were visualised and quantified using a LI-COR Odyssey^®^ imaging system. The expected molecular masses of the analysed proteins (without potential post-translational modifications) were checked using the Protein Information Resource Website (http://pir.georgetown.edu/ (accessed on 29 November 2022).

### 2.4. RNA Extraction and RT-qPCR

Total RNA was purified using RNeasy Plus reagents (Qiagen, Hilden, Germany), according to the manufacturer’s instructions. RT-qPCRs were performed with QuantiFast SYBR Green reagents and QuantiTect primer assays (Qiagen) using an Applied Biosystems 7500 Fast Real-Time PCR System (Waltham, MA, USA). Data were analysed with the software MxPro version 4 (Stratagene, San Diego, CA, USA). Genes of interest were normalised to *Ppia*. Fold changes were calculated using the 2^−ΔΔCt^ method [43].

### 2.5. Filipin Staining

Cells were grown in 96-well plates for 3 days. Filipin staining was performed with the Abcam cell-based cholesterol assay kit (ab133116). Images were obtained using a Carl Zeiss™ Axio Vert.A1 FL-LED Inverted Microscope (Zeiss, Oberkochen, Germany).

### 2.6. Immunofluorescence Microscopy

Cells cultured on VWR^TM^ coverslips (Sigma) were fixed with 4% paraformaldehyde (Thermofisher, Waltham, MA, USA) for 15 min on ice, permeabilised with 0.2% TritonX-100 (Sigma) for 10 min at room temperature, and incubated with 10% goat serum (Sigma) for 1 h at RT. Immuno-labelling with primary and secondary antibodies was performed for 1 h each in a humid environment and in the dark. Antibodies were diluted in blocking buffer to the final working concentrations, as stated in the above section. After incubation with DAPI (Sigma) diluted (1:20,000) in PBS and 0.1% Tween^TM^ 20, coverslips were mounted on slides using Dako Fluorescence mounting medium (Sigma).

Confocal images were acquired using a Zeiss LSM880 laser scanning microscope equipped with a 63X PlanApo 1.4 NA objective, GaAsP, and a Zeiss Airyscan 2 detector (Zeiss). Secondary-antibody-only controls were used to set threshold values. Z-stacks and maximum intensity images were generated using Zeiss Efficient Navigation (ZEN) 2.3 SP1 software. When comparing fluorescence intensities, images were taken with identical excitation and detection settings. The co-occurrence of N-SREBP2 staining with PML labelling was quantified using Fiji software (version 1.53f51 [44]) with the JACoP plugin [45]. Co-occurrence was determined based on the distance between geometrical centres, setting the threshold value for N-SREBP2 to 16. To test the robustness of our results, the analysis was carried out with a threshold of 32. The qualitative result was the same.

The In Situ Proximity Ligation Assay was performed with the Duolink kit (Sigma) following the manufacturer’s protocol with no modifications. The N-Srebp2 antibody was used in combination with the PML antibody. After incubation with primary antibodies, cells were incubated with oligonucleotide-conjugated anti-mouse minus and anti-rabbit plus proximity ligation assay secondary probes. Imaging was performed with a Zeiss LSM880 laser scanning microscope, and positive signals were visualised using ImageJ software version 1.52o.

### 2.7. Chromatin Immunoprecipitation (ChIP)

ChIP was performed using a ChIP-IT High Sensitivity kit (Active Motif, Waterloo, Belgium). Chromatin was digested to a size of 150–1000 base pairs using Micrococcal nuclease (New England Biolabs, Hitchin, Hertfordshire, UK). Fragmented chromatin was incubated overnight at 4 °C with the anti-N-SREBP2 antibody. Following immunoprecipitation, DNA cross-links were reversed, the proteins removed by Proteinase K, and the DNA recovered, purified, and quantified. RT-qPCR was performed using a QuantiTect SYBR Green PCR Kit (Qiagen) to detect the enrichment of N-SREBP2 at conserved *Hmgcr* sterol regulatory element-binding protein motif sequences [46]. The primer sequences were F: 5′CCAATAGGAAGGCCGCGATG3′ and R: 5′TCACGAACGGTCGCCCTAAC 3′. C_T_ values for the negative control ChIP samples (no N-SREBP2 antibody) were deducted from C_T_ values obtained from the SREBP2-ChIP samples, as they represent the background for the experiment.

### 2.8. Statistical Analyses

All data are reported as means ± SEM. Graphs were produced and analysed by GraphPad Prism software version 9.5.0. When three or more parameters were compared, an analysis of variance (ANOVA) was used. For the analysis of two groups, Student’s *t*-test was performed.

## 3. Results

### 3.1. SREBP2 Is Constitutively Processed to N-SREBP2

To explore the temporal, spatial, physiological, and quantitative relationships between SREBP2 and HMGCR regulation in the liver, we used the rat hepatoma McArdle-RH7777 cell line, which exhibits key aspects of hepatocyte function, including the production of VLDL [47,48], LDLR-mediated clearance of LDL-C [49], and ABCA1-mediated cholesterol efflux [50]. These cells were first incubated in a basal medium and then switched to a standard cholesterol-depleted (5% LPDS) medium with or without the synthetic, HMGCR-competitive inhibitor atorvastatin [51], following the same regime used to study sterol-responsive mechanisms of N-SREBP2 production in non-hepatic cells [33,52,53]. However, the mevalonate supplement was deliberately omitted to mitigate the potential effects of this endogenously produced HMGCR reaction product on HMGCR expression. Additionally, we first assessed the effect of sterol depletion on the protein levels of C-SREBP2 rather than on N-SREBP2, with the expectation of seeing a rise in the level of this SREBP2 cleavage product.

Transferring McArdle-RH7777 cells to the lipoprotein-depleted medium induced a transient rise in both HMGCR protein and RNA levels as early as 2 h after the switch (Figure 1A,B), while C-SREBP2 protein levels held steady throughout the 4 h incubation period (Figure 1A). The return of HMGCR protein (1.114 ± 0.067-fold, *p* = 0.449 for a difference from 0 h) and RNA (1.514 ± 0.500-fold, *p* = 0.445 for a difference from 0 h) levels towards basal levels at the 4 h time point was associated with a significant drop in *Ldlr* RNA (Figure 1B). We predicted that switching cells to the lipoprotein-depleted medium containing atorvastatin would produce a stronger effect on HMGCR expression compared to cells switched to the lipoprotein-depleted medium minus this statin. However, no such effect was observed over a 4 h incubation period (Appendix A). Additionally, atorvastatin did not increase C-SREBP2 protein levels (Appendix A).

Corroborating the data in Figure 1A that lipoprotein depletion induces a transient rise in HMGCR expression, no rise in HMGCR protein levels was observed in McArdle-RH7777 cells switched to the lipoprotein-depleted medium for 12 h (Figure 1C) and 24 h (Appendix A). By contrast, HMGCR, but not C-SREBP2, protein levels were higher in cells transferred to the lipoprotein-depleted medium plus the HMGCR-competitive inhibitor atorvastatin for these lengths of time (Figure 1C and Appendix A). The atorvastatin-induced rise in HMGCR protein levels (Figure 1C) was associated with an *Hmgcr/Ldlr* RNA profile (Figure 1D) that differed from that induced by lipoprotein depletion (Figure 1B). Collectively, these data indicate that the temporal, adaptative responses to sterol-poor conditions in McArdle-RH7777 and potentially other cells are governed by the sites of cholesterol depletion. We corroborated this interpretation using filipin, which marks cholesterol but not cholesteryl esters [54]. Under basal conditions, filipin strongly stained both plasma and intracellular membranes (Figure 1E, panel 1). After only 1 h in the lipoprotein-depleted medium, plasma membrane staining became visibly fainter, whilst that in intracellular bodies was more variable (Figure 1E, panel 2). After a further 3 h, these differences waned (Figure 1E, panel 3), congruent with the return of HMGCR RNA and protein levels to basal values (Figure 1A,B). By contrast, switching cells to the atorvastatin-containing medium produced a more homogeneous phenotype: 10 h following the switch, there was a whole-scale reduction in filipin staining of intracellular entities (Figure 1E**,** panel 5). This reduction was reversed by the 12 h time point (Figure 1E, panel 6), suggesting that this reversal was, at least in part, attributable to the rises in *Hmgcr* and *Ldlr* RNA (11 h time point, Figure 1D) and HMGCR protein (12 h time point, Figure 1C) levels that ensued in McArdle-RH7777 cells exposed to this inhibitor of HMGCR activity. However, similar to the observed restoration of cholesterol distribution in cells switched to the lipoprotein-depleted medium (Figure 1E, top panels), the recovery of cholesterol homeostasis in the statin-exposed cells could not be attributed to increased triggering of SREBP2 activation (0.853 ± 0.097-fold, *p* = 0.271 for a difference from 0 h, Figure 1C).

We next examined the subcellular distribution of the N-terminal fragment of SREBP2 (i.e., N-SREBP2) in McArdle-RH7777 cells cultured under basal conditions, expecting to see immunocytochemical staining of the nucleus and Golgi apparatus based on two observations. First, previous in vivo studies have shown that cell lysates from both murine [55,56,57,58] and human [59,60] liver samples contained substantial amounts of the N-SREBP2 fragment. Second, Western blot analyses returned comparable signals for the abundance of the C-terminal SREBP2 fragment in McArdle-RH7777 cells cultured under basal and cholesterol-depleted conditions (Figure 1A,C and Appendix A), indicating that, in these cells, the precursor SREBP2 polypeptide (Appendix A) is constitutively produced and transported to the Golgi apparatus for the release of the transcriptionally active N-terminal sequences. As shown in the representative images of pools of McArdle-RH7777 cells and the orthogonal projections of single cells, while the N-SREBP2 antibody strongly stained the nucleus (Figure 2A,B), staining of the ER, using a well-characterised KDEL monoclonal antibody [61] as a marker of the ER, was minimal (Figure 2A). Hence, these data suggest that, in these cells, the N-SREBP2 fragment is more abundant than the ER-bound precursor SREBP2 polypeptide from which it derives. Staining of the Golgi apparatus (Figure 2B) and of the cytoplasm (Figure 2A,B) was also observed, consistent with the pattern expected in cells trafficking the N-SREBP2 fragment to the nucleus [62]. In the nucleus, the staining was punctate and distributed in areas of relatively weak DAPI labelling, signifying that N-SREBP2 populated a transcriptionally active nuclear compartment(s) [63]. In comparison to the N-SREBP2 antibody, the N-SREBP1 antibody stained the nucleus (and ER) less conspicuously than the Golgi apparatus (Appendix A).

### 3.2. Nuclear Process Regulates N-SREBP2 Protein Levels

The distribution of N-SREBP2 in McArdle-RH7777 cells under basal conditions (Figure 2A,B) suggested that the rapid, >2.5-fold rise in *Hmgcr* RNA levels (Figure 1B) and the restoration of cholesterol homeostasis in these cells (Figure 1E) following their transfer to the lipoprotein-depleted medium might involve a nuclear, rather than an ER-cholesterol-sensing, mechanism. To investigate this, we compared changes in the abundance of N-SREBP2 and N-SREBP1 in the nuclei of McArdle-RH7777 cells switched to the lipoprotein-depleted medium for 1 h. Nuclear N-SREBP2 protein levels rose by 1.278 ± 0.009-fold (Figure 2C), while N-SREBP1 levels fell (Figure 2C), although we could not reliably quantify this reduction due to the relatively weak and rather diffuse immunoreactive signals obtained with the N-SREBP1 antibody in some Western blots. However, we note that a similar finding was found in the livers of mice administered ezetimibe to reduce intestinal cholesterol absorption, alongside the fungal-derived, relatively low-potency statin lovastatin [64]: specifically, while nuclear N-SREBP2 levels rose, those of the N-SREBP1 protein fell [65]. Here, we observed a fall in *Srebf1* RNA levels in McArdle-RH7777 cells switched to the lipoprotein-depleted medium for 2 h but no change in *Srefb2* RNA levels (Figure 2D). Additionally, no change in the abundance of *Srebf2* RNA was observed in cells switched to this medium for shorter periods (Figure 2D). Thus, these data, and the data in Figure 1A, indicate that increased SREBP2 expression is not responsible for the rapid rise in nuclear N-SREBP2 protein levels (Figure 2C), the near trebling of *Hmgcr* RNA levels (Figure 1A), or the restoration of cholesterol homeostasis (Figure 1E) that ensues in cells switched to the lipoprotein-depleted medium.

Since switching McArdle-RH7777 cells to the lipoprotein-depleted, atorvastatin-containing medium for 11 h increased *Hmgcr* RNA levels (Figure 1D), we checked whether transferring these cells to this medium for 10 h also induced a rise in nuclear N-SREBP2 protein levels. The analyses revealed that their nuclear fraction contained less, not more, of this transcription factor (Appendix A). Additionally, *Srebf2* RNA levels, which were significantly reduced at the 11 h time point (Appendix A), had returned to above basal levels at the 12 h time point (Appendix A). Thus, these findings and the Western blot analysis of the abundance of the C-terminal fragment of SREBP2 in cells switched to the atorvastatin-containing medium for 12 h (Figure 1C) suggest that the restoration of cholesterol homeostasis in cells exposed to this inhibitor of the ER-resident protein HMGCR (Figure 1E) may involve a mechanism that initially reduces, rather than increase, SREBP2 expression. By contrast, the data in Figure 1A,B,E and Figure 2C,D indicate that the immediate cellular response to a lack of an exogenous cholesterol supply involves a nuclear mechanism that increases nuclear N-SREBP2 protein abundance and *Hmgcr* RNA levels. Consistent with this view, switching McArdle-RH7777 cells to the lipoprotein-depleted medium increased the occupancy of N-SREBP2 at a conserved *Hmgcr* sterol regulatory binding element (Figure 2E). Thus, this finding, combined with other results reported herein (e.g., Figure 1A and Figure 2C,D) and those from an in vivo investigation [66] that found that labelled cholesterol delivered by oral gavage accumulated in rat liver nuclei and chromatin within 2 h, led to the following working hypothesis: a nuclear sterol switch regulates the stability of constitutively produced N-SREBP2 in hepatic cells.

To investigate the proposition that a cholesterol-sensitive nuclear process regulates nuclear N-SREBP2 turnover, we incubated McArdle-RH7777 cells under basal conditions in the presence of the translation inhibitor cycloheximide. A pilot showed that at the 4 h time point, when de novo SREBP2 synthesis and N-SREBP2 production from ER- and Golgi-bound SREBP2 (Figure 2A,B) were envisaged to make a minimal or no contribution to the nuclear pool of N-SREBP2, cell lysates contained ~40% less N-SREBP2 than at the 2 h time point (Figure 3A). We therefore adopted the 4 h time point to examine the effect of lipoprotein depletion on the disappearance of nuclear N-SREBP2 in the presence of cycloheximide (Figure 3B). Nuclear N-SREBP2 levels fell by 40.07 ± 8.22% and 48.48 ± 10.12% in cells incubated, respectively, for 4 h and 5 h with this protein translation inhibitor (Figure 3B). In cells incubated in the cycloheximide-containing medium for 4 h and then switched to the lipoprotein-depleted medium plus cycloheximide for a further 1 h, the corresponding figure was 40.99 ± 11.71% (*p* = 0.952 for a difference from incubation in a cycloheximide-containing medium for 4 h). Hence, relative to cells incubated in basal medium plus cycloheximide for 5 h, the cells incubated in the basal medium plus cycloheximide for 4 h and then switched to the lipoprotein-depleted medium (plus cycloheximide) for 1 h contained significantly more nuclear N-SREBP2 protein (Figure 3B). This finding, combined with the data in Figure 1B,E and Figure 2C–E and the in vivo studies of Erickson and colleagues [66], prompted us to further investigate the relationship between LPDS-induced cholesterol depletion, nuclear N-SREBP2 degradation, and *Hmgcr* RNA expression.

We first investigated whether promyelocytic leukaemia protein (PML) nuclear bodies might form part of a network involved in coordinating N-SREBP2 degradation and cholesterol homeostasis because previous studies have shown that these nuclear bodies modulate gene expression, for example, by sequestering SUMOylated proteins [67] and via their interactions with chromatin [68] and the ubiquitin–proteasome system [69,70]. With specific reference to N-SREBP2, three further findings seemed noteworthy. First, recombinant N-SREBP2-GFP, but not N-SREBP1-GFP, had been shown to accumulate in PML-sized nuclear structures within Hep2 (HeLa-derivative) cells [71]. Second, in fat-loaded Huh7 (hepatic) cells, PML staining was detected adjacent to nuclear lipid droplets [72], structures that comprise a cholesterol-rich monolayer and cholesteryl-ester-rich core. And, third, our inspection of two large-scale proteomic datasets revealed that N-SREBP2 is SUMOylated at amino acids K259 and K464 [70,73], consistent with the predictions for N-SREBP2 from a SUMO group-based system algorithm [74]. In fact, this tool also predicted that K249 and K420 reside in a SUMOylation motif. A resolvable PML and N-SREBP2 antibody combination was available to investigate the extent of the co-localisation of N-SREBP2 with nuclear PML bodies in human hepatoblastoma HepG2 [75], but not McArdle-RH7777, cells, and therefore, the potential contribution of nuclear PML bodies to the regulation of hepatic N-SREBP2 protein levels was examined in HepG2 cells. In the pilot analysis, the anti-PML antibody stained between 1 and 12 nuclear bodies per nucleus (5.923 ± 0.576), whereas the N-SREBP2 antibody stained numerous small foci (Figure 3C). The confocal images revealed the close association and/or co-localisation of N-SREBP2 with at least one PML nuclear body in ~50% of cells (Figure 3C,D). This closeness was verified by a proximity ligation assay that detects protein epitopes laying within 40 nm of each other (Figure 3E). In the follow-up experiment, we found that switching HepG2 cells to the lipoprotein-depleted medium for 1 h led to a small reduction in the number of PML nuclear bodies (4.772 ± 0.232 vs. 4.312 ± 0.272, *p* = 0.0341, Figure 3F) and the reduced co-occurrence of PML and N-SREBP2 staining: under basal conditions, N-SREBP2 staining overlapped with 1.955 ± 0.181 PML nuclear bodies/cell compared to 1.599 ± 0.16 under lipoprotein depletion (*p* = 0.0059 for difference, Figure 3G). These results, combined with previous findings [70,73], indicate that the sequestration of endogenously produced N-SREBP2 into PML nuclear bodies occurs at a relatively low level and that lipoprotein depletion reduces this accumulation. Hence, we next evaluated whether nuclear N-SREBP2 was a target of a nuclear ubiquitin–proteasome-mediated degradation pathway.

### 3.3. N-SREBP2 Restoring Cholesterol Homeostasis via Increased HMGCR Expression Escapes Ubiquitin–Proteasome-Mediated Degradation

We inspected large-scale mass spectrometry datasets [76,77,78] for evidence of N-SREBP2 ubiquitination. This identified 6 ubiquitinated lysine residues within this transcriptionally active fragment (Figure 4A): 4 within its 65-amino-acid-long HLHZ dimerisation motif (S344-I407) and 2 (K314, K464) within its flanking intrinsically disordered regions. The site of K314 ubiquitination seems doubly noteworthy; in other transcription factors, a disordered structure immediately before their basic DNA-binding sequences accelerates the recognition of cognate cis-acting gene promoter motifs [79]. On the other hand, the ubiquitination of such structures enhances their association with the proteasome and their subsequent degradation [80]. The ubiquitination of K464, which resides immediately downstream of the HLHZ dimerisation motif (Figure 4A), also seems notable, given that its SUMOylation [70] could serve to sequester undegraded N-SREBP2 into PML nuclear bodies.

Incubating McArdle-RH7777 cells with the proteasome inhibitor MG132 for 1 h produced the expected increase in nuclear ubiquitin-linked protein levels (Figure 4B) and a 1.582 ± 0.193-fold rise in nuclear N-SREBP2 (Figure 4B). These rises were not linked to higher *Srebf2* RNA levels (Figure 4C). Critically, the MG132-mediated rise in nuclear N-SREBP2 (Figure 4B) did not translate into significantly higher *Hmgcr* RNA levels (Figure 4D). For that, lipoprotein depletion was required (Figure 4D); then, the proteasome inhibitor increased *Hmgcr* RNA (2.391 ± 0.175-fold vs. 1.661 ± 0.062-fold, *p* = 0.0012 for a difference from the lipoprotein-depleted medium alone). Corroborating the experiment reported in Figure 1B, switching McArdle-RH7777 cells to the lipoprotein-depleted medium for 1 h did not increase *Ldlr* RNA (Figure 4E). This required the presence of MG132; then a significant, albeit modest (1.389 ± 0.087-fold, *p* = 0.0066 for a difference from the basal condition), rise was induced. Thus, these data, combined with the findings in Figure 2C, Figure 3B and Figure 4B, provide strong evidence that the ubiquitin–proteasome-mediated degradation of nuclear N-SREBP2 is significantly reduced under acute sterol depletion and that this reduced degradation is associated with its cholesterol-restoring transcription activities (Figure 4F). Thus, we propose that a nuclear ubiquitin–proteasome system and, to a lesser extent, PML nuclear bodies facilitate physiologically important pathways that prevent the pathological accumulation of the key transcription factor required for the delivery of a short-loop, organelle-based mechanism to maintain hepatocyte cholesterol homeostasis in an organ that handles continuous fluctuations in cholesterol supply and demand.

## 4. Discussion

Our nucleo-centric model of hepatic cholesterol homeostasis via the N-SREBP2-HMGCR axis (Figure 4F) unites previously unconnected biophysical and in vivo observations. One is the widely under-appreciated ability of cholesterol to interact with, and thereby increase the compactness of, chromatin [66,82], which seems highly pertinent given that the transcription of individual genes can be regulated by chromatin toggling between open (active) and compact states [83]. With specific respect to HMGCR, a key observation is that, in rats, the binding of cholesterol to liver chromatin peaks around the mid-point of their dark (feeding) phase and that this is followed by a rapid, very steep fall in HMGCR enzymatic activity [66]. The implication, which is consistent with the present in vitro studies, is that cholesterol sequestered by chromatin acts to suppress hepatic HMGCR expression, complementing the processes that regulate HMGCR protein turnover at the site of de novo cholesterol synthesis [84,85].

### The Significance of an Intranuclear Cholesterol-Signalling Mechanism for Regulating the SREBP2-HMGCR Axis

Tight regulation of the hepatic activity of the ER membrane protein HMGCR is essential for maintaining the liver’s many secretory functions, including supplying extrahepatic tissues with cholesterol to support their physiological functions. Here, we present strong evidence that cells producing VLDL, a process that involves loading ER-membrane-associated triglycerides, cholesterol, and cholesteryl esters onto apoB100 as it passes into the ER lumen [86], constitutively produce the transcriptionally active fragment of SREBP2 (i.e., N-SREBP2). Moreover, a sterol-signalling SREBP2-*HMGCR* axis that operates within the nucleus, rather than the SREBP2-sterol-sensing mechanism residing within the ER membranes of some extrahepatic cells [37,87], contributes to the control of hepatic cholesterol homeostasis.

Specifically, we propose that, in hepatocytes, the nuclear response to acute cholesterol shortage involves the promotion of *HMGCR* expression and the suppression of proteasome-mediated degradation of nuclear N-SREBP2 (Figure 4F). Our inference is that the proposed nuclear sterol-signalling SREBP2-*HMGCR* pathway works largely independently of the feedback system that controls cholesterol homeostasis via its regulation of HMGCR turnover at the site of cholesterol and non-sterol isoprenoid synthesis. In short, this system involves an ER-associated proteasome degradation pathway that targets HMGCR for degradation to protect against sterol accumulation in ER membranes [84,88,89].

We conjecture that the proposed nuclear control of cellular cholesterol homeostasis by the sterol-signalling N-SREBP2-*HMGCR* axis (Figure 4F) could operate in non-hepatic cells, including those that face challenging demands on their metabolism of cholesterol. In hepatocytes, the control of this axis, as well as that of proteasome-mediated degradation of HMGCR [84,88,89], is envisaged to reflect their unique and physiologically important role of loading large and dynamically varying amounts of dietary and de novo-synthesised cholesterol, cholesteryl esters, and triglycerides onto nascent apoB100 for delivery to extrahepatic tissues [90]. Our thinking, which seems consistent with the evidence of the constitutive production of hepatic N-SREBP2 (Figure 1A and Figure 2A,B,D), is that co-translationally loading apoB100 with cholesterol and cholesteryl esters as it enters the ER lumen [91] lowers ER membrane cholesterol sufficiently to enable the transport of precursor SREBP2 to the Golgi apparatus for the subsequent release of its transcriptionally active fragment to proceed constitutively.

We envisage that future studies will almost certainly connect the hepatocyte’s specialised control mechanisms of cholesterol homeostasis, including those mediated by the SREBP2-*HMGCR* pathway operating within the nucleus, to those exerted by Aster proteins [92] operating outside the nucleus [93]. In this regard, data indicate that this family of integral ER membrane proteins traffic excess cholesterol from the plasma membrane to the ER [94]. In fact, we note that the in vivo study of Tontonoz and co-workers [93] revealed that, in the liver, the hepatic Aster proteins (A and C) promote plasma membrane–ER cholesterol movement during prolonged (>>4 h) fasting and its subsequent removal from ER membranes via the production of VLDL. Moreover, when the authors examined the livers of mice following a 16 h fast, they found that those from the wild-type mice contained less HMGCR RNA and protein than those with the hepatic-specific knock-out of both Aster-A and -C, data indicating that that the HMGCR-ASTER axis is an important control point for regulating hepatic cholesterol homeostasis during prolonged fasting. With respect to SREBP2-mediated regulation of hepatic cholesterol homeostasis, the authors astutely commented that despite the finding that precursor SREBP2 protein levels in the livers from the Aster-A and -C knock-out mice fed a statin for 5 days were comparable to those in the equivalently treated wild-type mice, the nuclear fraction of the knock-out mice contained significantly more N-SREBP2—prompting the thought, consistent with the model proposed in this study (Figure 4F), that this may relate to the reduced nuclear degradation of N-SREBP2 under conditions of reduced cholesterol supply.

## 5. Conclusions

Our model proposes that N-SREBP2 and the nuclear ubiquitin–proteasome system form the missing response element between hepatic nuclear cholesterol availability and hepatic *HMGCR* expression (Figure 4F). A particular feature of the model is that structures that promote N-SREBP2 binding to gene promoters overlap with those that promote its degradation when levels of nuclear cholesterol (and as-yet unexplored metabolites) rise. Another is that the ubiquitination-mediated degradation of N-SREBP2 expelled from, or unable to re-occupy, the *HMGCR* promoter due to cholesterol-mediated chromatin compaction could rapidly amplify and accelerate the negative-feedback signal of rising cellular cholesterol to a physiological homeostatic advantage.

There are several limitations in this study that we would like addressed in future studies. These include quantifying changes in nuclear cholesterol levels upon cholesterol-depletion and the effects of these changes on local chromatin compaction across the genome in cells that produce apoB lipoproteins, as well as cells that have high rates of cholesterol utilisation. Then, correlating these data with changes in N-SREBP2-mediated *HMGCR* transcription and estimates of de novo cholesterol synthesis using labelled substrates [95,96]. Complementary Ultraperformance Liquid Chromatography–Electrospray Ionisation–Mass Spectrometry analyses would also be desirable for establishing changes in the subcellular compartmentalisation and concentrations of the HMGCR substrate HMG-CoA, its reaction product, mevalonate, and mevalonate pathway intermediates that contribute to the restoration of cellular cholesterol homeostasis following acute cholesterol depletion. In particular, we note that HMG-CoA may be distributed in three different intracellular compartments (cytosol, mitochondria, and peroxisomes) [97]. Additionally, we note that the turnover of the cytosolic HMG-CoA synthase (HMGCS1) that provides this substrate to HMGCR for mevalonate synthesis has recently been shown to be upregulated during prolonged nutritional stress [98].

With respect to the technical difficulties and expense of generating genome-wide-scale genomic data and targeted metabolomic datasets from primary hepatocytes under multiple conditions, we note that both the McArdle-RH7777 and HepG2 cancer cell lines used in the current study have previously provided a robust platform for obtaining such data [48,99], as well as key mechanistic insights into processes regulating hepatic cholesterol homeostasis [47,100,101].

In future studies, it would be informative to define the precise N-SREBP2 amino acids involved in mediating the balance between gene promoter binding and ubiquitin–proteasome-mediated degradation under basal and cholesterol-depleted conditions in both McArdle-RH7777 and HepG2 cells, as well as primary hepatocytes, including those with a reduced ability to transport the accessible pool of plasma membrane cholesterol to the ER [93]. Additionally, given recent data that indicate that the C-terminal SREBP2 fragment is ubiquitinated at multiple sites (Appendix A), we envisage that determining the importance of each of these sites for precursor and C-SREBP2 turnover would help define the extent of the cross-talk between the processes regulating the ER-associated proteasome-mediated degradation of HMGCR and the ASTER-mediated transfer of plasma membrane-accessible cholesterol to the ER with those regulating the nuclear N-SREBP2-HMGCR axis. Arguably, a more immediate aim would be to identify the specific nuclear E3 ubiquitin ligase(s), deubiquitinase(s), and SUMO proteins regulating nuclear N-SREBP2 turnover, as we envisage that this would accelerate establishing the therapeutic potential of modulating nuclear N-SREBP2 protein levels in common diseases using in vivo disease models.

## Figures and Tables

**Figure 1 cells-13-01255-f001:**
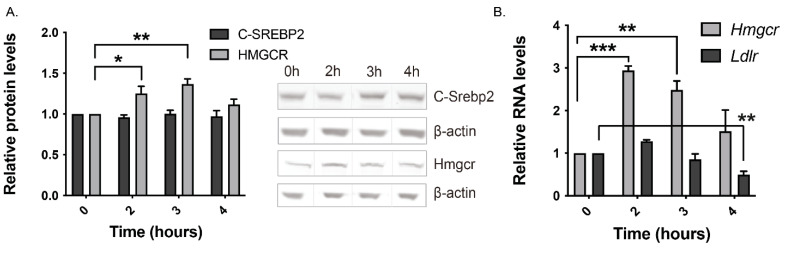
HMGCR, but not C-SREBP2, protein is increased by sterol depletion. (**A**,**C**) Western blot analysis of C-SREBP2 and HMGCR in whole-cell lysates of McArdle-RH7777 cells switched from the basal medium (0 h) to a medium containing lipoprotein-depleted serum (LPDS) for the indicated time points (**A**) and LPDS plus 1 μM atorvastatin for 12 h (**C**). The C-terminal SREBP2 fragment was detected with an anti-C-SREBP2 antibody raised against SREBP2 amino acid residues 794–1141. β-Actin was used as the loading control. Black line separators show non-contiguous lanes from the same gel. (**B**,**D**) RT-qPCR quantifications of specified transcripts are normalised to the house-keeping gene *Ppia.* In D, cells were switched to LPDS plus 1 μM atorvastatin for 11 h. (**A**–**D**) Data (mean ± SEM) expressed relative to basal levels are from five independent experiments, each with three technical replicates. Statistical differences from 0 h were determined by one-way ANOVA followed by Dunnet’s multiple-comparison test. Significance is indicated by asterisks: *, *p* < 0.05; ** *p* < 0.01; *** *p* < 0.001. (**E**) Filipin staining of McArdle-RH7777 cells cultured under basal conditions and switched to the LPDS-containing medium (top row, lanes 2–3) and the LPDS-containing medium plus 1 μM atorvastatin (bottom row, lanes 2–3) for specified times. Images are representative of two independent experiments with similar results.

**Figure 2 cells-13-01255-f002:**
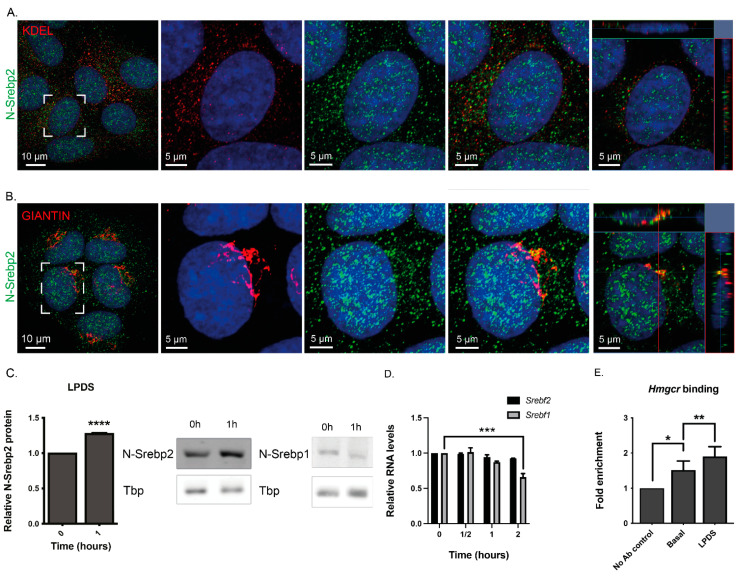
Lipoprotein depletion increases the nuclear abundance of the constitutively produced N-SREBP2 fragment and its occupancy at the *Hmgcr* promoter. (**A**,**B**) N-SREBP2 immunoreactive sequences reside predominately in the nucleus. McArdle-RH7777 cells cultured under basal conditions were subject to indirect immunofluorescence using specified antibodies and counterstained with DAPI. Left-hand panels show maximum intensity projection images of representative fields of cells. Right-hand panels show expanded views of the regions highlighted by white squares. Enlargements show the maximum intensity images derived from collating specified signals from all stacks of the indicated cells (middle panels) and orthogonal views (right-hand-side panels) of the middle stack (XZ axis, right-hand border; YZ, top border) sliced through the centre of the nucleus. N-SREBP2 (green), the ER marker KDEL (red), and the Golgi marker Giantin (red). (**C**) The quantification of N-SREBP2 protein levels in nuclear extracts prepared from McArdle-RH7777 cells cultured under basal conditions and cells switched to the lipoprotein-depleted medium (LPDS) for 1 h, plus representative immunoblots. The Tata-box binding protein (TBP) is the loading control. Normalised data (mean ± SEM), expressed relative to basal conditions, are from four independent experiments, each with three technical replicates. **** *p* < 0.0001 versus cells at 0 h, determined by an unpaired two-tail Student’s *t*-test. (**D**) RT-qPCR quantification of *Srebf1* and *Srebf2* in total RNA of McArdle-RH7777 cells switched to the lipoprotein-depleted medium (LPDS) for specified time points. Data (mean ± SEM), normalised to *Ppia*, are from three independent experiments, each with three technical replicates. Statistical differences from 0 h were determined by one-way ANOVA followed by Dunnet’s multiple-comparison test. *** *p* < 0.001. (**E**) Chromatin immunoprecipitation–qPCR indicated that switching McArdle-RH7777 cells to the lipoprotein-depleted medium (LPDS) for 1.5 h was associated with increased N-SREBP2 binding at a conserved *Hmgcr* sterol regulatory binding element. Data (mean ± SEM) from three independent experiments, each with three technical replicates, are represented as fold enrichment over the no-antibody (Ab) control. Statistical differences were determined by the Student’s *t*-test. Asterisks indicate significant occupancy of N-SREBP2 to the conserved *Hmgcr* sterol regulatory element under basal conditions and increased occupancy of this transcription factor to this element in lipoprotein-depleted (LPDS) medium. * and **, *p* < 0.05 and 0.01.

**Figure 3 cells-13-01255-f003:**
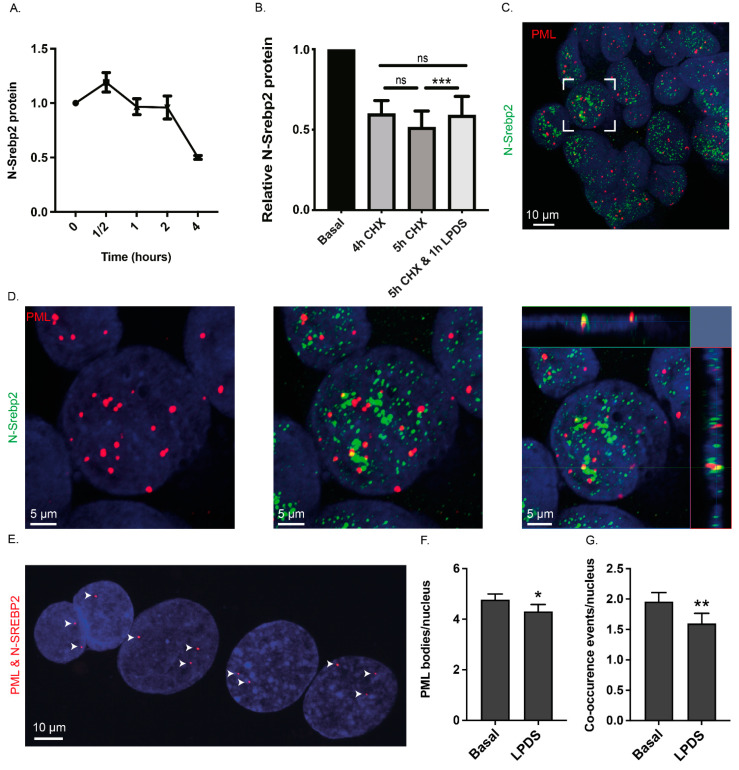
Sterol depletion protects nuclear N-SREBP2 from degradation and sequestration into PML nuclear bodies. (**A**) Western Blot analysis of N-SREBP2 in total cell lysates prepared from McArdle-RH7777 cells incubated in the basal medium containing 10 μM cycloheximide (CHX) for specified durations. Data (mean ± SEM) for this pilot analysis are from two independent experiments, each containing three technical replicates. (**B**) Western Blot analysis of N-SREBP2 in nuclear extracts prepared from McArdle-RH7777 cells incubated in a 10 μM CHX-containing medium for specified durations. At the 4 h time point, one set of cells was switched to the lipoprotein-depleted medium (LPDS) for 1 h (5 h CHX + 1 h LPDS): CHX was present throughout the 5 h experiment. Data (mean ± SEM) expressed relative to no CHX are from three independent experiments, each with three technical replicates. Significance between 5 h CHX + 1 h LPDS versus 5 h CHX was determined by an unpaired one-tailed Student’s *t*-test. ***, *p* < 0.001; ns: no significant difference. (**C**,**D**) The co-occurrence of N-SREBP2 with PML nuclear bodies. HepG2 cells cultured under basal conditions were fixed and subject to indirect immunofluorescence using PML and N-SREBP2 antibodies. Cells were counterstained with DAPI. An expanded view of the region marked by a white square (**D**) shows the separate channels (panels 1, 2) and orthogonal views (XZ axis, right-hand side; YZ, top) of merged images through the centre of the nucleus. (**E**) Proximity ligation assay images of DAPI-stained nuclei overlaid with the PLA dots (arrowheads) generated by the same antibody pairings used in (**C**,**D**). (**F**) Fewer PML nuclear bodies in HepG2 cells switched to the lipoprotein-depleted medium (LPDS) for 1 h. (**G**) The co-occurrence of N-SREBP2 labelling with PML nuclear body staining in the same nuclei analysed in F, irrespective of PML numbers. For (**F**,**G**), the data (mean ± SEM) are from 250 cells per independent experiment (n = 6) for each experimental condition. Significance was determined by a one-tailed paired Student’s *t*-test: * and **, *p* < 0.05 and 0.01, for a difference from the basal condition.

**Figure 4 cells-13-01255-f004:**
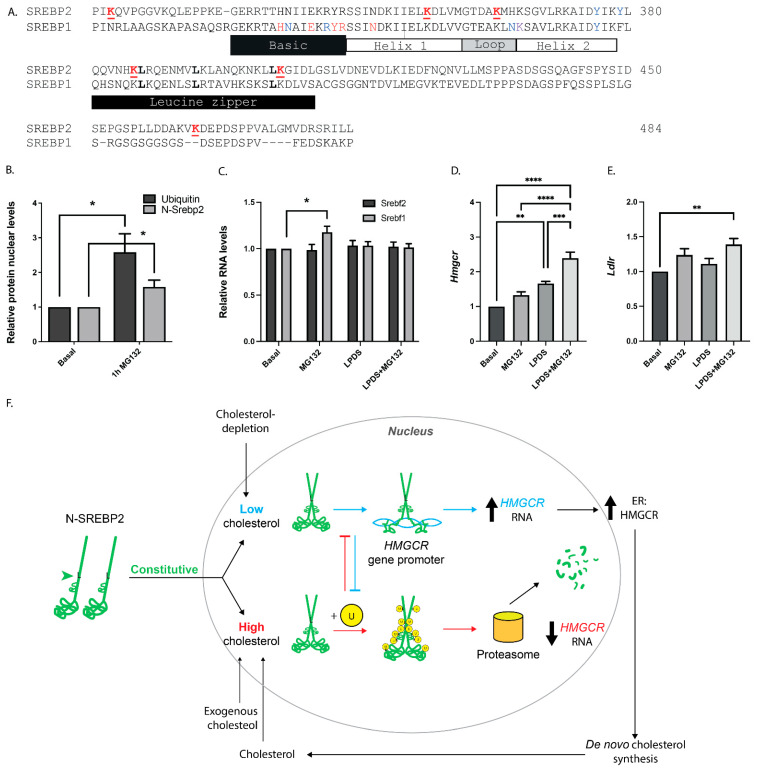
Inhibition of nuclear N-SREBP2 proteasome-mediated degradation amplifies the impact of sterol depletion on *HMGCR* expression. (**A**) Sites of N-SREBP2 ubiquitination (red, underlined). Data extracted from ubiquitinomes of HEK293T cells [76,77,78]. The secondary structure and amino acids (red) making contacts with *SREBF1* promoter sequences inferred from co-crystal structures of the SREBP1-bHLHZ fragment bound to a 38-nucleotide oligonucleotide containing an asymmetric sterol regulatory element [81]. Leucine (L) residues of the leucine zipper highlighted in bold. (**B**) The proteasome inhibitor MG132 reduces nuclear N-SREBP2 degradation. McArdle-RH7777 cells cultured under basal conditions and with MG132 (10 μM) for 1 h. Nuclear fractions were immunoblotted with ubiquitin (black bars) and N-SREBP2 (grey bars). Three independent experiments, each containing three technical repeats, were performed. TBP was used as the loading control. Normalised data (mean ± SEM) are expressed relative to basal (no MG132) values. Significance was determined by Student’s *t*-test (basal ubiquitin vs. 1 h MG132 and basal N-SREBP2 vs. 1 h MG132). (**C**–**E**) RT-qPCR quantification of specified transcripts in McArdle-RH7777 cells incubated with MG132 for 1 h (column 2) or switched to the lipoprotein-depleted (LPDS) medium and LPDS + MG132 (10 μM) for 1 h. Data (mean ± SEM) from six independent experiments, each with three technical replicates. Data, normalised to *Ppia*, are expressed relative to basal values. Statistical differences were determined by one-way ANOVA: * = *p* < 0.05; ** *p* < 0.01, *** = *p* < 0.001, **** = *p* < 0.0001. (**F**) The proposed model. In hepatic cells, N-SREBP2 is constitutively produced and transported into the nucleus. Upon cholesterol depletion, N-SREBP2′s access to the *HMGCR* promoter is enhanced, increasing HMGCR expression and restoring cholesterol homeostasis. Under cholesterol-replete conditions, cholesterol binding to chromatin [66,82] reduces N-SREBP2′s access to the *HMGCR* promoter and becomes more susceptible to proteasome-mediated degradation. For clarity, N-SREBP2′s transactivating sequences are not depicted. U = ubiquitin. L = leucine zipper.

## Data Availability

Publicly available datasets were examined in this study. The datasets are described in the methods and results section.

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
