# Peer review of "N-SREBP2 Provides a Mechanism for Dynamic Control of Cellular Cholesterol Homeostasis"

_cells, 2024, doi:10.3390/cells13151255_

Round 1

Reviewer 1 Report

Comments and Suggestions for Authors

In the manuscript entitled “Intranuclear Sterol-regulated Degradation of Hepatic N- SREBP2 Provides a Mechanism for Dynamic Control of Cellu- lar Cholesterol Homeostasis”, Tozen Ozkan-Nikitaras et al.. The authors used RH7777 and HepG2 cell lines to demonstrate a new model of SREBP2 mediated regulation of cholesterol biosynthesis. The manuscript is well-written, and the topic is very intriguing. There are some minor comments:

1.   In Fig 1E, the authors observed reduction was reserved by the 12h time point, and the authors mentioned the reduction was consistent with the rises in HMGCR and Ldlr expression in Fig 1C,1D. The cells were given different treatment, it is inappropriate to make this conclusion.

2.   In Fig 2, there are some positive KDEL signal observed in nuclear, KDEL signal should not be found in nuclear. Please make sure the KDEL signal is real.

Author Response

In Fig 1E, the authors observed reduction was reserved by the 12h time point, and the authors mentioned the reduction was consistent with the rises in HMGCR and Ldlr expression in Fig 1C,1D. The cells were given different treatment, it is inappropriate to make this conclusion.

Thank you for carefully reading the manuscript. We have made two revisions to address this point. First in the text (page 5) and in the Figure legend. The cells were given the same 'treatment' - in figure 1E, two time points  were examined based on the data in Figure 1C, 1D. 

In Fig 2, there are some positive KDEL signal observed in nuclear, KDEL signal should not be found in nuclear. Please make sure the KDEL signal is real.

Thank you again, we have added two revisions to reaasure you that the KDEL signal is real. The first is included in the text (page 7). The second involves adding a better figure legend to explain the analysis of the confocal images. 

Reviewer 2 Report

Comments and Suggestions for Authors

The manuscript of Ozkan-Nikitaras et al. deals with the mechanistic aspects of cholesterol regulation in hepatic cells, mainly the involvement of the sterol regulatory element binding protein 2 (SREBP2) and the ER-resident protein HMG-CoA reductase (HMGCR), as regulators of cholesterol biosynthesis. Authors found that nuclear N-SREBP2 became resistant to proteasome-mediated degradation. This resistance was paired with increased occupancy at the HMGCR promoter and HMGCR expression. Moreover, inhibiting nuclear N-SREBP2 degradation did not increase levels of HMGCR RNA level after cholesterol-depletion. Authors suggest a new model of SREBP2-mediated regulation of cholesterol biosynthesis in the liver, where nuclear cholesterol and the ubiquitin-proteasome system provide a short-loop system that modulates the rate of cholesterol biosynthesis, maintaining cellular cholesterol homeostasis. This work appeared to be well performed, showing interesting data. However, I have some minor concerns:

 1)    The changes in the SREBP2 and the HMGCR, as found in the present study, are reflected in the capacity of the de novo cholesterol synthesis by the studied tumoral cells?    

 2)    Authors say that nuclear N-SREBP2 levels fell by 40.07 ± 8.22% and 48.48 ± 10.12% in cells incubated respectively for 4-h and 5-h with the protein translation inhibitor cycloheximide. Did they determine protein levels for N-SREBP1, as a comparison of the effect of protein synthesis inhibition?

 3)    Under the experimental conditions here reported, there were also    changes in synthesis of HMG-CoA which is mainly mitochondrial? 

4)    In the page 12, second paragraph, if the sentence “Significance of an intranuclear cholesterol-signaling mechanism for regulating the SREBP2-HMGCR axis” is a subtitle, it should write in italics.

Author Response

1)    The changes in the SREBP2 and the HMGCR, as found in the present study, are reflected in the capacity of the de novo cholesterol synthesis by the studied tumoral cells 

Thank you for your insightful comments, which should be considered in the design of future studies aimed at understanding the dynamics of the nuclear N-SREBP2-HMGCR axis in restoring cholesterol homeostasis during acute sterol-depletion/insufficiency. Hence, we have revised the second and third paragraph of the conclusion to briefly comment on the actual type of analyses that could be used to produce this understanding.  Surprisingly, we could not find a study that specifically looked at the effect of HMGCR overexpression on de novo cholesterol synthesis in the cells we used.  Chronic overexpression of endogenous HMGCR has been described in two non-hepatic tumour cell lines; in both, these were associated with increased de novo cholesterol synthesis but the mechanism triggering the overexpression of HMGCR in these cell linbes was not explored. 

2. Authors say that nuclear N-SREBP2 levels fell by 40.07 ± 8.22% and 48.48 ± 10.12% in cells incubated respectively for 4-h and 5-h with the protein translation inhibitor cycloheximide. Did they determine protein levels for N-SREBP1, as a comparison of the effect of protein synthesis inhibition?

We did not examine the impact of cycloheximide on the disappearance of nuclear N-SREBP1 because acute sterol-depletion did not lead to higher nuclear N-SREBP1 protein levels. Only nuclear N-SREBP2 protein levels were increased by acute sterol-depletion and in the setting of no change in SREBP2 expression or evidence of increased processing of the precursor SREBP2 polypeptide.  In comparison,  our RT-qPCR data indicated that changes in the expression of Srebf1 RNA and the subsequent processing of the encoded product (i.e. precursor SREBP1 polypeptide) may affect (or be affected) nuclear N-SREBP1 protein levels. 

3).  Under the experimental conditions here reported, there were also changes in synthesis of HMG-CoA which is mainly mitochondrial? 

Thank you for this question, which we have addressed in the revised manuscript (in the limitation part of the Conclusion). We acknowledge this should be considered in future metabolomic studies designed to address the importance of the nuclear N-SREBP2-HMGCR axis in restoring cellular cholesterol homeostasis under condition of acute sterol insuffiency. We remind readers that the HMGCR substrate HMG-CoA resides in both the mitochondria and the cytosol subcellular compartments of the cell. Moreover, turnover of the cytosolic form of the enzyme producing this substrate for utilisation on the melavonate pathway is markedly increased under conditions of prolonged nutritional stress. 

4). In the page 12, second paragraph, if the sentence “Significance of an intranuclear cholesterol-signaling mechanism for regulating the SREBP2-HMGCR axis” is a subtitle, it should write in italics.

Corrected.